# An In Vitro Evaluation of Primary Stability Values for Two Differently Designed Implants to Suit Immediate Loading in Very Soft Bone

**DOI:** 10.3390/dj9010005

**Published:** 2021-01-08

**Authors:** Dirk Krischik, Selen Ergin Tokgöz, Andreas van Orten, Anton Friedmann, Hakan Bilhan

**Affiliations:** 1Private Dental Practice Zahnärzte Do24, Dortmunder Str. 24–28, 45731 Waltrop, Germany; dirk.krischik@t-online.de (D.K.); andreas.van.orten@icloud.com (A.v.O.); 2Private Dental Practice, Dentaglobal Oral Health Centre, Adalet Str 2131/10 No:3A Bayraklı, İzmir 35530, Turkey; 3Department of Periodontology, School of Dentistry, Faculty of Health, Witten/Herdecke University, Alfred-Herrhausen Str. 44, 58455 Witten, Germany; anton.friedmann@uni-wh.de (A.F.); hakanbilhan@gmail.com (H.B.)

**Keywords:** primary stability, type 4 bone, thread design, immediate loading, resonance frequency analysis

## Abstract

The achievement of sufficient implant stability in poor quality bone seems to be a challenge. Most manufacturers develop special dental implants, which are claimed to show higher stability even in very soft bone. The aim of this experimental study was to compare two recently introduced dental implants with differing thread designs. A total of 11 implants of each group were inserted in the part of the fresh bovine ribs, corresponding to very soft bone. The primary stability was measured with resonance frequency analysis (RFA) and Periotest; the average of two measurements for each method and for each implant was taken and statistical analysis was applied. The highest stability values were obtained with the ICX Active Master implants, followed by the Conelog^®^ Progressive-Line implants placed with the very soft bone protocol. The primary stability values of the Conelog^®^ Progressive-Line implants inserted by the very soft bone protocol and the ICX Active Master implants placed with the standard protocol showed sufficient stability for immediate loading in low-density bone. Within the limitations of this study, the thread design of the implants and underdimensioned implant bed preparation seem to be effective for better primary stability in cancellous bone.

## 1. Introduction

Dental implants have been used as a reliable and successful treatment modality for decades. The indication variety is broad, but the immediate loading has become very popular in the latest years. The maxilla especially presents a real challenge with its very often encountered low density bone. Both the implant’s length and diameter, together with the bone quality, have been reported as factors important for primary stability [1]. To comply with this purpose, manufacturers are seeking an implant design capable of achieving high stability values during placement even in a very poor-quality bone. Accounting for the latter, the manufacturers are investing efforts into optimization of the implant design in the sense of enhanced primary stability. Primary stability is obtained by mechanical retention and friction in the bone, which relates to the geometry of the implant besides the surgical skills of the clinician [2,3]. Additionally, the adherence with the drilling protocol seems to play an important role in achieving a congruent implant bed, thus increasing the friction and retention. If an immediate loading is planned, an insertion torque over 30 Ncm and an implant stability quotient (ISQ) exceeding 60 resonance frequency analysis (RFA) units is strongly recommended for obtaining sufficient mechanical stability. The rationale behind is the reported tolerance of the bone to micromovements ranging between 50–150 µm [4]. The micromovements are known to be detrimental to implant integration range above 150 µm as a fibrotic encapsulation may be their result instead of osseointegration, ending up in implant loss [5]. The immediate loading may be considered a valid therapeutic choice even for low-density bone if at least 45 N/cm of insertion torque are ascertained [6]. The recommended implant stability quotient (ISQ) score is supposed to exceed 60–65 [7,8,9]. These numbers determine the boundaries of the risk threshold; once surpassed, an immediate loading of just inserted implant may be considered feasible. Accounting for the Periotest assessment, the negative Periotest values below 0 appear to fit the above-mentioned threshold.

Recently two manufacturers introduced specifically designed implants claimed to be suitable for immediate loading in low-density bone. The purpose of this study was to test the primary stability by comparing the ISQ scores and the Periotest values obtained in vitro for two these implants placing both according to the drilling protocols recommended by the manufacturers. 

### Null Hypothesis

The null hypothesis was that there would be no statistically significant differences in primary stability values for different thread designs and implant drilling protocols.

## 2. Materials and Methods 

This study was implemented as an in vitro study. The sample was composed of fresh bovine ribs which were suitable for imitating the soft bone. The narrow part of the rib was supposed to match bone type 4 at closest accordance to the Lekholm and Zarb classification and selected for the study procedures (Figure 1) [10,11].

The predictor variables were two different implant designs proposed by two manufacturers and the primary outcome variables were three different drilling protocol for comparison of the primary stability in vitro:

A: The ICX Active Master implant (Medentis Medical, Walporzheim, Germany) with 3.75 mm diameter and 12.5 mm length operated according to the standard drilling protocol for soft bone (Figure 2). 

B1: Conelog^®^ Progressive-Line implant (Fa. Camlog GmbH, Wimsheim, Germany) (Figure 3) with 3.8 mm diameter and 11 mm length operated according to original drilling protocol for soft bone.

B2: Same implant as for the group B1 operated according to the drilling protocol for very soft bone aiming at underdimensioned preparation for improving implant stability.

A total of 11 implants was calculated to be sufficient for each group (see below). The implant bed preparations were performed in the narrow side of bovine ribs respecting the recommended safe distance of 3 mm between two adjacent units by the same clinician (Figure 4).

The implant bed preparation strictly followed the manufacturer’s recommendations; each implant was inserted with its rough area completely covered by crestal bone. Healing abutments, 2 mm in height, were screwed into the implants immediately after insertion. The stability measurements were completed by one of the investigators, blinded to the type of the implant and the drilling protocol tested. The electronic percussive testing tool was used first (Periotest M, Medizintechnik Gulden, Modautal, Germany). 

The hand piece of the Periotest device was positioned perpendicular to the healing abutment of each implant. Two measurements in an angle of 90 degrees to each other were performed for each implant and the average was taken as one value. After the percussion test, the RFA values were assessed once the magnetic pegs were installed on each implant by turns instead of the gingiva formers (MEGA ISQ Implant Stability—Measurement Device, Megagen, Montagnola, Switzerland). The probe of the Osstell device was positioned at 1 mm apart from the peg in a 90-degree angle (Figure 4). The RFA values resulted in ISQ scores with a range from 1 to 100 (with 100 implying the highest degree of stability). 

The measurements were carried out in duplicates for each implant positioning the probe tip from two different directions, and the average of the ISQ scores was recorded as the value per implant [12].

### Statistical Analysis

The methodology was reviewed and approved by an independent statistician. The mean and standard error were calculated for quantitative variables. All statistical analyses were performed by IBM SPSS Statistics 25 (IBM SPSS, Version 25.0, Armonk, New York, United States). The sample size for each group was determined by G-power analyses. A sample size of 11 implants was considered suitable with 80% power for each RFA group and with 98% for each Periotest group using a two-sided, Chi-square hypothesis test with a significance level (alpha) of 0.05. The normal distribution of the data was tested by conducting a Shaprio-Wilk test. The results from both the RFA and Periotest assessments did not show the normal distribution. Therefore, the Kruskal-Wallis test was used for inter-group comparisons as the normal distribution was discarded. The Mann-Whitney U test was performed to test the significance of pairwise differences using Bonferroni adjustment for the multiple comparisons. The level of significance was set at *p* = 0.05. Correlation between the RFA and Periotest values per drilling protocol group allocation (A; B1; B2) were assessed by Spearman’s correlation test (2-tailed). The level of significance was set at *p* = 0.01.

## 3. Results

The mean primary stability values for all experimentally inserted implants are shown in Table 1. 

The primary stability in the groups showed statistically significant differences as for the RFA data (*p* = 0.018) as for the Periotest numbers (*p* = 0.003). While the group A implants disclosed highest primary stability scores (RFA: 71.39 ± 8.92 and Periotest: −5.45 ± 1.69), for the group B1 the lowest primary stability scores were obtained (RFA: 61.93 ± 4.33 and Periotest: −1.7 ± 1.92). The mean primary stability scores for group B2, inserted using the “very soft bone protocol”, were 68.55 ± 6.67 for RFA and −3.84 ± 1.91 for Periotest, respectively (α = 0.05). 

The paired comparisons of ISQ scores were shown in Table 2 and *p* value was set as 0.017 according to Bonferroni correction. 

Statistically significant greater ISQ scores were obtained with the implants from group A compared to the implants placed with the drilling protocol for type 3–4 bone (soft bone) from group B1 and group B2 (*p* = 0.016). The group B2 implants placed with the drilling protocol for very soft bone showed no statistically significant ISQ scores compared to the group B1 implants placed with the drilling protocol for type 3–4 bone (soft bone) (*p* = 0.018; *p* > 0.017). Although mean ISQ values of the group A implants exceeded those for the group B2 implants, the difference was not statistically significant (*p* = 0.438).

The paired comparisons of Periotest scores were shown in Table 3 and *p* value was set as 0.017 according to Bonferroni correction.

Statistically significant higher frequency of negative scores was obtained with the group A implants compared to group B1 implants (*p* = 0.002). The group B2 implants placed with the drilling protocol for very soft bone showed no statistically significant Periotest scores compared to the group B1 implants placed with the drilling protocol for type 3–4 bone (soft bone) (*p* = 0.033; *p* > 0.017). The mean Periotest scores for the group A implants showed greater negative numbers than group B2 implants placed with the drilling protocol for very soft bone, but the difference was not statistically significant (*p* = 0.082).

RFA and Periotest scores showed significantly high correlation for all three protocols. The correlation ratio for group A was calculated as −0.980; −0.916 for group B1; and −0.909 for group B2 (Table 4).

The null hypothesis was accepted by the results of the study, since the thread design as well as the drilling protocol had influence on the implant stability, but not at a statistically significant level.

## 4. Discussion

The purpose of the present in vitro study was to evaluate the potential of two different implant designs in achieving primary stability in very soft bone. Although the in vitro settings do not adequately reflect the true clinical environment, the data may be considered helpful in choosing the convenient implant geometry and type for the certain indication. Several experimental studies indicated that stability and bone to implant contact, respectively, will be increased by implant design with a smaller pitch, higher number of threads, deeper threads, and decreased thread helix angle. The additional implant length and/or the greater diameter were also proved beneficial in this regard [13,14,15]. Bovine ribs were chosen to mimic the in vivo low-density bone as used in various previous studies [16,17,18]. 

Several methods have been proposed to measure the implant stability, among which RFA and electronic percussive testing (EPT) have broad clinical acceptance [12,19]. In a recent study, it was pointed out that implant stability measurements would not always allow for differentiation of implants varying in shape and placed with different protocols and compressive testing of bone was recommended for prediction of implant performance [20]. Nevertheless, implant stability measurements provide objective and repeatable data so that clinically and experimentally reliable conclusions may be drawn. For this reason, both methods are considered very suitable for obtaining data to make comparisons [18].

This study revealed a highly significant correlation between RFA and EPT scores (Table 4) per group. This observation corroborated the results reported from previous studies [21,22]. The modified drilling protocol for the type B implants recommended an underdimensioned preparation of the implant bed and differed hereby for group B2 from the standard protocol used for group B1 and group A implants. In the present study, this modification in the drilling protocol was helpful in achieving comparable stability values for group B2 implants to those for group A. The underdimensioned implant bed preparation was already reported being associated with an increased stability in a poor-quality bone [3,16,23,24]. The comparison of both drilling protocols for the type B implants revealed the superiority of underdimensioned preparation in achieving statistically significant higher stability values. However, the non-modified drilling protocol applied with group A implants resulted in somewhat greater scores for stability than displayed by group B-2 (Figure 2). Although the mean Periotest scores for the Group A showed more negative scores than the Group B2, indicating a trend in greater stability, the difference was not statistically significant (*p* = 0.082) (Table 3). 

As all implants were placed in an assumable similar bone quality by the same operator, the rationale for the different stability was supposed to be related to the design of the implants. Considering the outcome with the standard protocol for the group B-1, the results indicated that the achieved stability was not feasible for immediate loading (Table 1). Therefore, the recommendation to use the modified drilling protocol together with the implant type B should be extended to either soft bone quality in general. This will release the practitioner from deciding which protocol is to be preferred as the discrimination between soft or very soft bone is pertinent and the implementation of immediate load in low-density bone should be evaluated with care [25]. 

There are a few studies pointing out to importance of square shaped threads in implants for achieving higher implant stability [26]. Thread height as well as the thread width appeared important [3,27]. The reason for statistically significant higher stability values with the group A implants may be the above mentioned square shaped thread design, obviously resulting in a greater thread height than in the implants allocated to both groups B1 and B2. 

This study only considered the mechanical features and the primary implant stability. Neither the osseointegration, peri-implant tissue response, nor biological or mechanical complications could be extrapolated from this data. This study was performed in very soft bone and no interpretation is possible concerning the behavior after bone condensing with osteotomes or in bone types with higher bone density.

Within the limitations of the study, implants with a square shaped thread design and bone condensing property appeared rather to justify better primary stability in very soft bone, thus being feasible for immediate loading. The second implant group offered the prospective to achieve in vitro sufficient stability to be used for immediate load by applying a special underdimensioned drilling protocol.

## 5. Conclusions

The results of this study suggest that the thread design of the implants as well as underdimensioned implant bed preparation are important for increasing implant stability, so that even in very soft bone safe stability values for immediate loading can be reached. Future studies might be conducted to focus on the different primary stability values for immediate loading in vivo experiments. Future studies may be designed as prospective cohort study in order to determine lowest and safest primary stability values on type 4 and even very soft bone.

## Figures and Tables

**Figure 1 dentistry-09-00005-f001:**
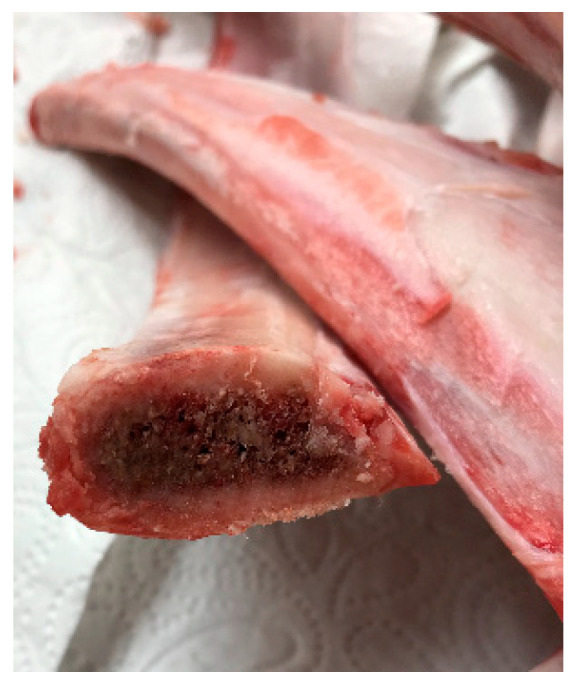
The part of the ribs which was similar to type 4 was used in the study.

**Figure 2 dentistry-09-00005-f002:**
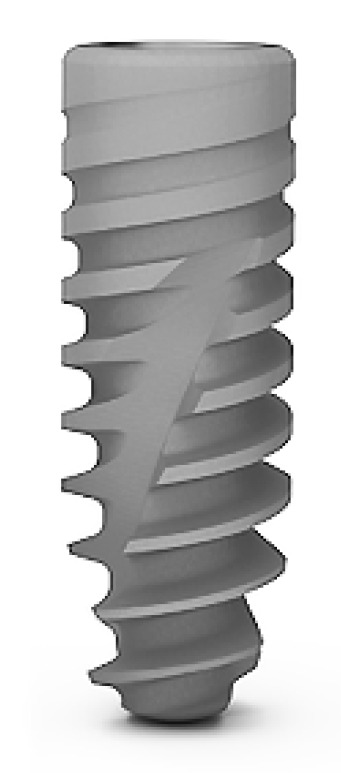
ICX Active Master implant.

**Figure 3 dentistry-09-00005-f003:**
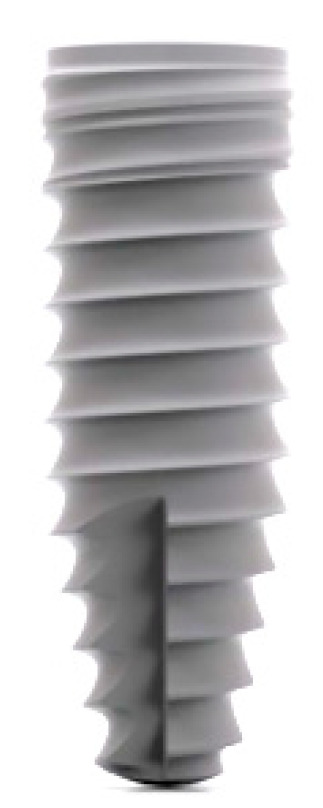
Conelog^®^ Progressive Line implant.

**Figure 4 dentistry-09-00005-f004:**
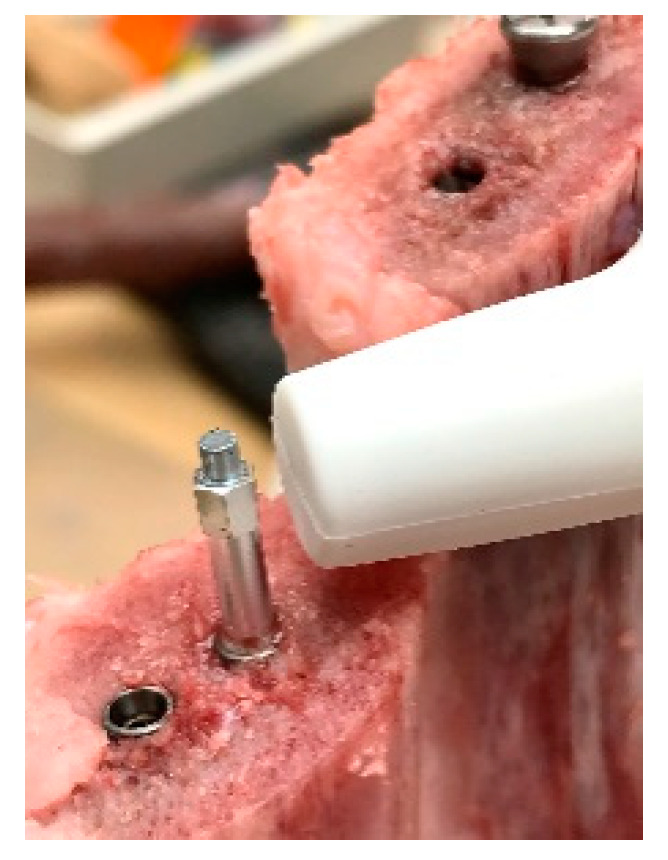
For each protocol implants were placed into the selected bovine ribs with a safe distance to each other by the same clinician. The probe of the Osstell Mentor was held 1 mm from the peg at a 90-degree angle.

**Table 1 dentistry-09-00005-t001:** Statistical evaluation of the mean stability values of each drilling protocol with standard deviation.

	Group A Mean ± SD	Group B1 Mean ± SD	Group B2 Mean ± SD	*p*
RFA	71.39 ± 8.92	61.93 ± 4.33	68.55 ± 6.67	0.0181
PERIOTEST	−5.45 ± 1.69	−1.7 ± 1.92	−3.84 ± 1.91	0.0031

Kruskall-Wallis test. SD: Standard deviation, RFA: Resonance frequency analysis.

**Table 2 dentistry-09-00005-t002:** Mean and standard deviation of RFA values in paired comparison for all three protocols.

	Group A Mean ± SD 71.39 ± 8.92	Group B1 Mean ± SD 61.93 ± 4.33	Group B2 Mean ± SD 68.55 ± 6.67
Group A-*p* value	-	0.0161	0.438
Group B1-*p* value	0.0161	-	0.0181
Group B2-Pvalue	0.438	0.0181	-

Mann-Whitney U test (2-tailed).

**Table 3 dentistry-09-00005-t003:** Mean and standard deviation of Periotest values in paired comparison for all three protocols.

	Group A Mean ± SD −5.45 ± 1.69	Group B1 Mean ± SD −1.7 ± 1.92	Group B2 Mean ± SD −3.84 ± 1.91
Group A-*p* value	-	0.0021	0.082
Group B1-*p* value	0.0021	-	0.0331
Group B2-*p* value	0.082	0.0331	-

Mann-Whitney U test (2-tailed).

**Table 4 dentistry-09-00005-t004:** Correlation between RFA and Periotest values for all three protocols.

	Group A Periotest		Group B1 Periotest		Group B2 Periotest	
	*p*	r	*p*	r	*p*	r
Group A RFA	0.00	−0.980 *				
Group B1 RFA			0.00	−0.916 *		
Group B2 RFA					0.00	−0.909 *

Spearman correlation test (2-tailed).

## Data Availability

The data presented in this study are not publicly available due to experimental nature of the study.

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
