# Peer review of "An In Vitro Evaluation of Primary Stability Values for Two Differently Designed Implants to Suit Immediate Loading in Very Soft Bone"

_dentistry, 2021, doi:10.3390/dj9010005_

Round 1

Reviewer 1 Report

Please see the enclosed file

Author Response

Reviewer 1:

Review of the manuscript entitled “An In Vitro Evaluation of Primary Stability Values 2 for Two Differently Designed Implants to Suit 3 Immediate Loading in Very Soft Bone” (TD: dentistry- 1045716), submitted to Dentistry Journal.

This manuscript proposes an in vitro study on the effect of the implant design as well as of the drilling protocol on the primary stability of dental implants. The proposed method and the analysis of results were clearly presented, leading some interesting preliminary conclusions which may help clinicians to improve the the primary stability of dental implants. However, before publishing it is necessary to address several points as follows:

  • Table 4, which was supposed to present the correlation between the RFA and Periotest results, is missed. In the manuscript, the Table 3 and Table 4 are identical.

Response to Reviewer 1: sorry for this inconvenience, Table 4 was added now (replaced by the original table 4).

  • Can the authors better explain why the effect of the drilling protocol for very soft bone was not studied for The ICX Active Master implant (group A) ? 

Response to Reviewer 1: We have taken the specifications of the manufacturers; the “very soft bone drilling protocol” was only given by Conelog implants.

Reviewer 2 Report

Remarks and recommendations:

Article title: The last part of the title - in particular the term "very soft bone"(row four) is better to be replaced by something more appropriate: for example "bone quality type IV"

Introduction:

Rows: 63-65 – The Null hypothesis is clear defined, but It’s declared into the section - “Introduction”. It would be better if a subsection or a new one is created - for example: Aim of the study.

Materials and Methods:

Rows 122-123 – the declared usage of Bonferroni correction coefficient and finally accepted p-value of .05 is inaccurately. I will explain in a little more details.

Example (concerning the current article) : If the Bonferroni adjustment for the multiple comparisons is used, the new p-value will be the alpha-value (α-original =0.05) divided by the number of comparisons [in our case three (3) declared groups – A, B1 and B2], or: (α-altered = 0.05/3) = 0.017 (0.01666666). To determine if any of the three (3) correlations is statistically significant, the p-value must be p< .017*.

_______________________

*If you are unsure about this, please consult above with specialist of bio-statistics.

Unfortunately, the declared level of significance after the correction with the Bonferroni coefficient remains 0.05.

NB. The name & position of specialist of Bio-statistics not mentioned.

Results:

NB. Important:

The data of table three and table four are the same. The data of Table four not correspond with declared data of the rows 161 and 162.

NB. It's remained unclear the standpoint of the authors about declared Null Hypothesis into the rows 63-65. It must be stated precisely: are the data, provided by statistical analysis allow to confirm or reject the Null hypothesis. This must be included into the end of section “Results”.

NB. The data of declared usage of Spearman’s correlation test (rows 124 & 125) are not presented anywhere in the "Results" section.

Discussion:

Row 165 – “The purpose of the present in vitro experiment was to evaluate …” the word “experiment” is better to change with “study”, “analysis”, etc.

Rows 211-218 The whole paragraph is out of the topic discussed. Moreover, the cited opinion about applying high insertion torque “… of 110.6 to 176 N/cm ...” does not in agreement with the applied torque during implant placement, recommended by almost all manufacturers dental implant systems. A reference can be made here (Straumann surgical procedures catalogue): https://www.straumann.com/content/dam/media-center/straumann/smart/com/en/smart-multi/clinical-theory-e-books/490.093-SmartM-2-1-com-en.pdf. “ … page 36 of the Straumann’s surgical instruction last paragraph (with red letters), states: "An insertion torque of 35 Ncm is recommended to place the implants." For the reasons mentioned above, I believe that the whole paragraph should be dropped (rows 211-218).

Author Response

Reviewer 2:

Remarks and recommendations:

Article title: The last part of the title - in particular the term "very soft bone"(row four) is better to be replaced by something more appropriate: for example "bone quality type IV“

Response to Reviewer 2: The authors have chosen the term “very soft bone” intentionally in order to differentiate between type IV and even more cancellous bone, as stated by one of the manufacturers.  However, we would like to follow the advice and we will change the title if the reviewer insists.

Introduction:

Rows: 63-65 – The Null hypothesis is clear defined, but It’s declared into the section - “Introduction”. It would be better if a subsection or a new one is created - for example: Aim of the study. 

Response to Reviewer 2: Done. We have added the following contents:

"Aim of the study:

The null hypothesis was that there would be no statistically significant differences in primary stability values for different thread designs and implant drilling protocols."

Materials and Methods: 

Rows 122-123 – the declared usage of Bonferroni correction coefficient and finally accepted p-value of .05 is inaccurately. I will explain in a little more details.

Example (concerning the current article): If the Bonferroni adjustment for the multiple comparisons is used, the new p-value will be the alpha-value (α-original =0.05) divided by the number of comparisons [in our case three (3) declared groups – A, B1 and B2], or: (α-altered =0.05/3= 0.017 (0.01666666). To determine if any of the three (3) correlations is statistically significant, the p-value must be p< .017*.

_______________________

*If you are unsure about this, please consult above with specialist of bio-statistics.

Unfortunately, the declared level of significance after the correction with the Bonferroni coefficient remains 0.05. 

Response to Reviewer 2: Sorry for this inconvenience, statistical analysis was revised according to Bonferroni correction. The results were reanalyzed according to the new p values. The results, discussion and conclusion parts were rearranged slightly according to the new results of the statistical analysis.

 The name & position of specialist of Bio-statistics not mentioned.

Response to Reviewer 2: The statistical analysis was performed by co-author S.E.T. (DDS, PhD, Private Dental Practice, Prosthodontist, Dentaglobal Oral Health Centre).

Results:

  1. Important:

The data of table three and table four are the same. The data of Table four not correspond with declared data of the rows 161 and 162. 

Response to Reviewer 2: Sorry for this inconvenience; table 4 was replaced by the original one which was by mistake mixed up with table 3 (table 3 was taken twice). We have corrected the tables.

  1. It's remained unclear the standpoint of the authors about declared Null Hypothesis into the rows 63-65. It must be stated precisely: are the data, provided by statistical analysis allow to confirm or reject the Null hypothesis. This must be included into the end of section “Results”.

Response to Reviewer 2: A statement was added to the end of the results section. 

 NB.

The data of declared usage of Spearman’s correlation test (rows 124 & 125) are not presented anywhere in the "Results" section.

Response to Reviewer 2: The results of the Spearman’s correlation test (row 166-168) are now presented in the text and the new Table 4 was inserted, too.

Discussion:

Row 165 – “The purpose of the present in vitro experiment was to evaluate …” the word “experiment” is better to change with “study”, “analysis”, etc. 

Response to Reviewer 2: Thank you for your suggestions. We have revised it.

Rows 211-218 The whole paragraph is out of the topic discussed. Moreover, the cited opinion about applying high insertion torque “… of 110.6 to 176 N/cm ...” does not in agreement with the applied torque during implant placement, recommended by almost all manufacturers dental implant systems. A reference can be made here (Straumann surgical procedures catalogue):  https://www.straumann.com/content/dam/media-center/straumann/smart/com/en/smart-multi/clinical-theory-e-books/490.093-SmartM-2-1-com-en.pdf. “ … page 36 of the Straumann’s surgical instruction last paragraph (with red letters), states: "An insertion torque of 35 Ncm is recommended to place the implants." For the reasons mentioned above, I believe that the whole paragraph should be dropped (rows 211-218). 

Response to Reviewer 2: The whole paragraph was removed from the discussion and also the references 28 and 29 from the reference list, since it was out of topic indeed.

Reviewer 3 Report

dear authors, this study evaluate the stability values of two different implant as an in vitro study.

the study is interesting for the readers and well performed. I suggest the publication in the journal.

Author Response

Reviewer 3:

dear authors, this study evaluate the stability values of two different implant as an in vitro study. The study is interesting for the readers and well performed. I suggest the publication in the journal.

Thank you

Round 2

Reviewer 2 Report

After the corrections made, I believe that the quality of the article has significantly improved.